# Physical localization of 45S rDNA in *Cymbopogon* and the analysis of differential distribution of rDNA in homologous chromosomes of *Cymbopogon winterianus*

Shivangi Thakur[1☯], Upendra Kumar[2☯], Rashmi Malik[3], Darshana Bisht[1], Priyanka Balyan[4], Reyazul Rouf Mir[5], Sundip Kumar[1] *

1 Department of Molecular Biology and Genetic Engineering, College of Basic Sciences and Humanities, G. B. Pant University of Agriculture & Technology, Pantnagar, India, 2 Department of Molecular Biology, Biotechnology and Bioinformatics, CCS Haryana Agricultural University, Hisar, India, 3 Department of Genetics and Plant Breeding, College of Agriculture, G. B. Pant University of Agriculture & Technology, Pantnagar, India, 4 Department of Botany, Deva Nagri P.G. College, CCS University Meerut, Meerut, India, 5 Division of Genetics & Plant Breeding, Sher-e-Kashmir University of Agricultural Sciences & Technology of Kashmir (SKUAST-Kashmir), Srinagar, (J&K), India

☯ These authors contributed equally to this work.
* malik.sundip@gmail.com

**Data Availability Statement:** All relevant data are within the manuscript.

## Abstract

*Cymbopogon*, commonly known as lemon grass, is one of the most important aromatic grasses having therapeutic and medicinal values. FISH signals on somatic chromosome spreads off *Cymbopogon* species indicated the localization of 45S rDNA on the terminal region of short arms of a chromosome pair. A considerable interspecific variation in the intensity of 45S rDNA hybridization signals was observed in the cultivars of *Cymbopogon winterianus* and *Cymbopogon flexuosus*. Furthermore, in all the varieties of *C. winterianus* namely Bio-13, Manjari and Medini, a differential distribution of 45S rDNA was observed in a heterologous pair of chromosomes 1. The development of *C. winterianus* var. Manjari through gamma radiation may be responsible for breakage of fragile rDNA site from one of the chromosomes of this heterologous chromosome pair. While, in other two varieties of *C. winterianus* (Bio-13 and Medini), this variability may be because of evolutionary speciation due to natural cross among two species of *Cymbopogon* which was fixed through clonal propagation. However, in both the situations these changes were fixed by vegetative method of propagation which is general mode of reproduction in the case of *C. winterianus*.

## Introduction

*Cymbopogon*, commonly known as lemon grass, is one of the most important aromatic grasses belonging to family *Poaceae* with proven therapeutic and medicinal values. It is grown for commercial and industrial purposes in tropics and subtropics of Asia, America and Africa [1]. Industrial interest in citronella oils is due to extensive use of its various components as fragrance in perfumes, soaps, mosquito repellent and as flavour additives in food products. *Cymbopogon* species display wide variation in morphological attributes and essential oil

**Funding:** The author(s) received no specific funding for this work.

**Competing interests:** The authors have declared that no competing interests exist.

composition at inter and intraspecific levels [2]. The first cytological study in Indian *Cymbopogon* to ascertain the chromosome number [3, 4] indicated the different ploidy levels in these genera varying from diploid (2n = 20) to tetraploid (2n = 40) and hexaploid (2n = 60). However, as per available reports in literature, the cytogenetic studies in this genus have been limited to chromosome count and preliminary karyotype description of the cultivated species. The characterization of *Cymbopogon* germplasm largely been done on phenotypic characteristics [5] and more recently on the basis of some molecular markers such as RAPD and SSR [6–8]. Analysing the genome organization of plants reveal evolutionary relationships of different genomes, which may also be useful for crop improvement. The ribosomal RNA genes represent two highly conserved tandemly arrayed gene families namely 45S rDNA and 5S rDNA which have been studied extensively in plant genomes. Because of the numerous copies of these highly conserved families of repeated sequences, their physical location on the chromosome can be easily visualized. The 45S rDNA sites in somatic chromosomes are most extensively utilized and widely documented chromosomal regions in eukaryotes through fluorescent *in situ* hybridization (FISH). The 45S rDNA family together with the intergenic spacer (IGS) is present as tandem arrays within the nucleolus organizer regions (NORs) of satellite chromosomes and also at other chromosomal sites where they may not be associated with NOR [9–11]. Length polymorphism of these repeat units has been reported in a wide range of plants and animals and are attributed to variation in number of sub-repeats that are found in IGS. The length of IGS and the chromosomal location of 45S rDNA genes are often characteristic of a species and were widely used to study the phylogenetic relationships of several plant species [12–15]. In view of the limited cytological reports on *Cymbopogon*, molecular cytogenetic studies for the identification of individual chromosomes are urgently needed in this important crop. Therefore, the present investigation was conducted using 45S rDNA as a probe to develop valuable FISH landmarks of somatic chromosomes of *Cymbopogon* which may be utilized in subsequent molecular cytogenetic studies to generate physical maps of *Cymbopogon* species.

## Results

### Karyotype analysis

All the four varieties studied during the present investigation were observed as diploid with chromosome number (2n = 20). Variety Bio-13 belonging to *C. winterianus* is one of the supreme significantly commercialized variety of Java Citronella grass. This variety of *C. winterianus* contain diploid chromosome complement as 2n = 20 with basic chromosome number x = 10 (Figs 1a, 1c and 2a). Arm ratios for the chromosomes of this species ranged from 1.05 to 3.37 (Fig 2a) and the range of chromosome lengths lies between 1.3 to 3.12 μm. Bio-13 observed to have 3 chromosomes in the range of 2.0 to 3.0 μm (3C) and rest of the seven chromosomes in the range of 1 to 2 μm (7D). As per the arm ratios of different chromosomes of this variety, 3 chromosomes were metacentric and 7 were sub-metacentric (Table 1).

Manjari (*C. winterianus*) had showed its somatic chromosome complement as 2n = 20 (diploid) with basic chromosome number x = 10 (Fig 1d) similar to Bio-13. The arm ratios as well as range of chromosome lengths (Fig 2b) was almost equivalent to *C. winterianus* var. Bio-13. This variety is observed to have 2 chromosomes in the range of 2.0 to 3.0 μm (2 C chromosomes) and rest of the eight chromosomes ranged in 1.0 to 2.0 μm (8 D chromosomes). As per the arm ratios (Table 1) of different chromosomes of this variety, 3 chromosomes were metacentric and 7 were sub-metacentric.

Medini (*C. winterinaus*) is comparatively new variety than above two varieties of *C. winterianus* which is also classified under Java Citronella grass with 2n = 20 as its somatic

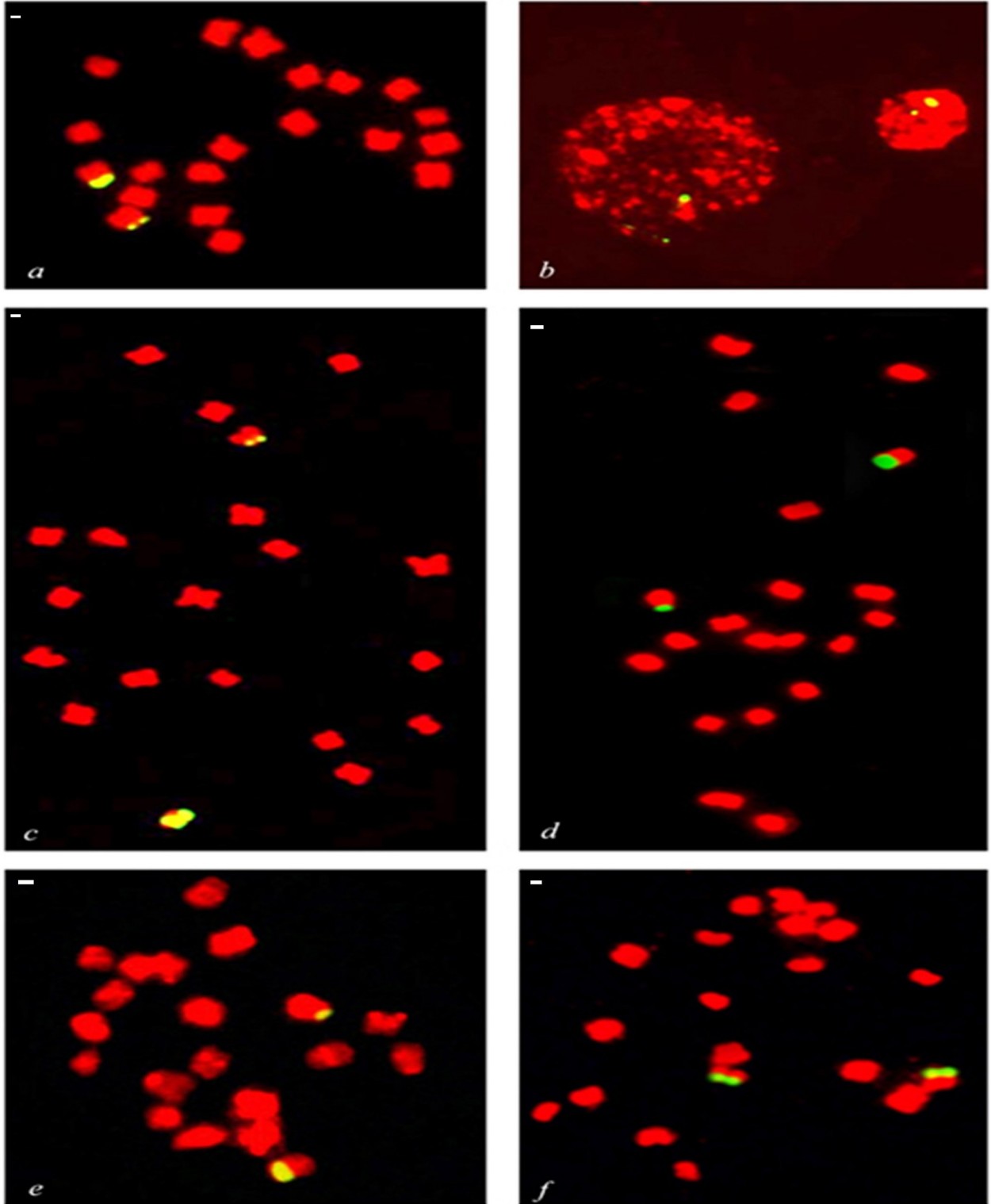

**Fig 1. Mitotic cells of variety Bio-13, Medini, Manjiri (*C. winterianus*) and Krishna (*C. flexuosus*) after fluorescence *in situ* hybridization with 45S rDNA (yellow/green) as FISH probe.** The chromosomes were counterstained with propidium iodide (red). **(a and c)** Somatic metaphase showing diploid chromosomes of Bio-13 ($2n = 20$) showing differential hybridization signals of 45S rDNA on a pair of somatic chromosomes. **(b)** Interphase nuclei of Bio-13 showing differential hybridization signals of 45S rDNA **(d)** Somatic metaphase showing diploid chromosomes of Manjari ($2n = 20$) showing differential hybridization signals of 45S rDNA on a pair of somatic chromosomes. **(e)** Somatic metaphase showing diploid

chromosomes of Medini (2*n* = 20) showing differential hybridization signals of 45S rDNA on a pair of somatic chromosomes. (**f**) Somatic metaphase showing diploid chromosomes of Krishna (2*n* = 20) showing hybridization signals of 45S rDNA on a pair of somatic chromosomes. White bars on right side of every figure is equivalent to 1 μm.

chromosome complement (Fig 1e). Arm ratios for the chromosomes of this species ranged from 1.04 to 3.40 and the range of chromosome lengths lies between 1.70 to 3.10 μm (Fig 2c). The length of two chromosome ranged between 2.0 to 3.0 μm (2 C chromosomes) and other eight chromosomes were ranged in 1.0 to 2.0 μm (8 D chromosomes). As per the arm ratios of different chromosomes of this variety, 3 of them were characterized as metacentric whereas 7 of them were characterized as sub-metacentric (Table 1).

Krishna (*C. flexuosus*), also contains the somatic chromosome complement of 2*n* = 20 (Fig 1f). The arm ratios of the chromosomes of this species ranged from 1.07 to 2.22 and the range of chromosomes length lies between 1.48 to 2.30 μm (Fig 2d). This showed that *C. flexuosus* var. Krishna had slightly smaller chromosomes than *C. winterianus*. As per the arm ratios of different chromosomes of this variety, 5 of the chromosomes were metacentric and other 5 were sub-metacentric (Table 1).

## Localization of 45S rDNA

Bio-13 (*C. winterianus*) had shown strong hybridization signal of 45S rDNA at the terminal ends of short arms of the longest heterologous pair of somatic chromosome complement. The differential signal intensity of 45S rDNA was observed in different stages like metaphase (Fig 1a and 1c) and interphase (Fig 1b). It was found that relative length of 45S rDNA hybridization signals were different on this particular chromosome pair. The data represented that 45S rDNA hybridization signals had covered 29.48% (0.29 μm) of whole chromosome while on another chromosome of this pair covered only 10.01% (0.28 μm). Interestingly, this difference in signal intensity of 45S rDNA hybridization signals on both the chromosomes of this heterologous pair confirmed differential copy number of 45S rDNA in this case.

Manjari (*C. winterianus*) had also shown strong hybridization signals of 45S rDNA at the terminal ends of short arms of longest heterologous pair of its chromosome complement (Fig 1d). Similar to the Bio-13, the differential signal intensity of 45S rDNA was observed in metaphase chromosomes of Manjari (Fig 1d). The relative length of 45S rDNA signal in one chromosome of the heterologous bivalent was 26.51% (0.79 μm), while the relative length of 45S rDNA in other chromosome of this pair was recorded 11.59% (0.32 μm) which is indicating the differential copy number of rDNA in both chromosomes of this heterologous chromosome pair.

Medini (*C. winterianus*) had shown to have strong hybridization signals of 45S rDNA at the terminal ends of short arms of longest heterologous pair of its chromosome complement (Fig 1e). Similar to the Bio-13 and Manjari, this variety of *C. winterianus* also showed differential signal intensity of 45S rDNA in metaphase chromosomes. The relative length of 45S rDNA signal in one chromosome of the heterologous bivalent was 26.12% (0.81 μm) while the relative length of 45S rDNA in other chromosome of this pair was recorded 11.72% (0.32 μm) which is indicating the differential copy number of rDNA in both chromosomes of this heterologous chromosome pair.

Krishna (*C. flexuosus*) had shown to have strong hybridization signals of 45S rDNA at the terminal ends of short arms of longest chromosome pair of its chromosome complement. However, in contrast to differential signal intensity of 45S rDNA shown in all the three varieties of *C. winterianus*, this species had shown to have the 45S rDNA signals of similar intensity i.e., 32.60% and 33.18% on both chromosomes of longest homologous chromosome pair (Fig 1f).

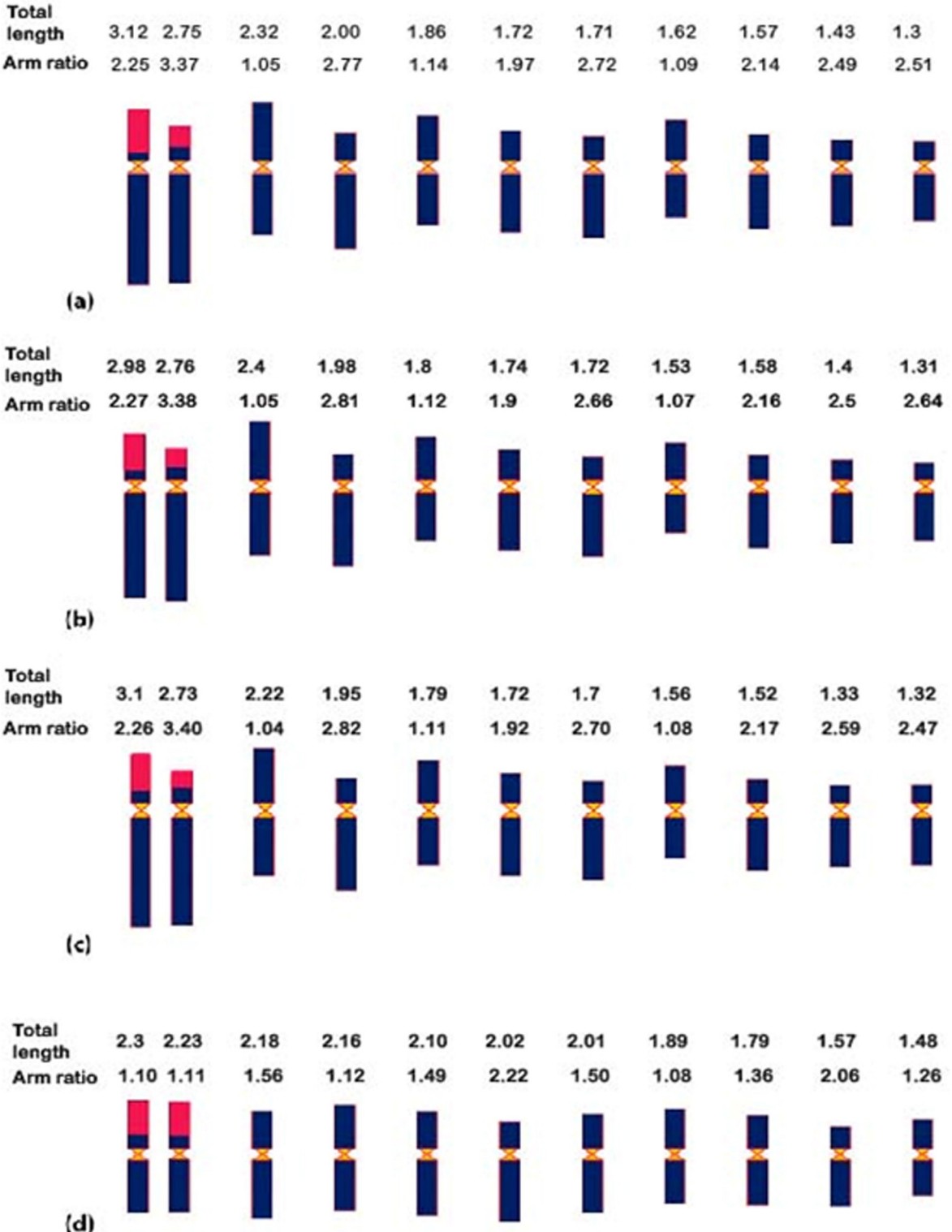

**Fig 2. Total length and arm ratio of chromosome.** Schematic representation of chromosome arranged in descending order of their length: **a**. Karyogram of Bio-13 (*C. winterianus*), **b**. Karyogram of Manjari (*C. winterianus*), **c**. Karyogram of Medini (*C. winterianus*), **d**. Karyogram of Krishna (*C. flexuosus*).

**Table 1. Description of karyotyping in all the varieties.**

| Variety | Number of cells studied | Longest chromosome* (µm) | Shortest chromosome* (µm) | Arm ratio* | rDNA length | rDNA % | No. of sub-metacentric chromosome (Sm) | No. of metacentric chromosome (M) | Range of length (Chromosomes) | |
|---|---|---|---|---|---|---|---|---|---|---|
| | | | | | | | | | 2–3 µm | 1–2 µm |
| Bio-13 | 40 | 3.12 | 1.30 | 1.05–3.37 | 0.28 | 29.48 | 7 | 3 | 3 | 7 |
| | | | | | 0.92 | 10.01 | | | | |
| Manjari | 40 | 2.98 | 1.31 | 1.05–3.38 | 0.32 | 26.51 | 7 | 3 | 2 | 8 |
| | | | | | 0.79 | 11.59 | | | | |
| Medini | 40 | 3.10 | 1.70 | 1.04–3.40 | 0.32 | 26.12 | 7 | 3 | 2 | 8 |
| | | | | | 0.81 | 11.72 | | | | |
| Krishna | 40 | 2.30 | 1.48 | 1.07–2.22 | 0.74 | 32.88 | 5 | 5 | 6 | 4 |
| | | | | | 0.75 | 33.28 | | | | |

*Note: Average values.

## Discussion

The observed results on chromosome length and arm ratios for chromosome of *C. winterianus* (Fig 2) were found to be in almost accordance with previous karyomorphological observation [16]. As per the arm ratios of different chromosomes of all the genotypes of *C. winterianus*, 3 chromosomes are metacentric and 7 are sub-metacentric. These results are in contrast to the previous investigation [16] where *C. winterianus* was reported to have 8 sub-metacentric and 2 metacentric chromosomes. This difference may be due to small size of the chromosomes as sometimes it may be difficult to identify the centromeric position of the chromosomes. However, during the present investigation we have recorded the data on several cells and the data were based on average length and hence seems to be more authentic. The slight variation observed in the karyotypes of *C. winterianus* may be due to phenomenon of differential chromatin condensation in different chromosomes. Although, lemon grass variety Krishna (*C. flexuosus*) has shown its somatic complement as $2n = 20$ with basic chromosome number $x = 10$ (Figs 1f and 2d) similar to other species of *Cymbopogon* but slight variation is noted in average chromosome ratio of *C. flexuosus* (1.95) than earlier reports [14] of arm ratios (1.53).

In the past few decades, the extensive study of many plant species for localization of 45S rDNA genes have been done through FISH technique and it has been observed that, most of diploid plants have two sites rDNA i.e., a single locus [17]. Even though, some diploids exist with numerous sites of rDNA. The rDNA copy number changes rapidly and frequently, triggering relocation or deletion of some loci over and above the reduction in copy number below the detection sensitivity limit or mapping resolution [18, 19]. For the justification of these differences some mechanisms have been proposed by various workers, such as various chromosome rearrangements, unequal crossing-over, gene transposition (gene mobility), and conversion [19–21]. There is certain proof that rDNA sites may alter chromosomal position, lacking the contribution of translocations [20], mobile rDNA tandem repeats with different molecular mechanisms [22, 23] which also includes triggering by *En/Spm* transposons [19, 22, 24, 25]. Examples for arithmetical changes in 45S rDNA sites were described for few additional species, like in the colchicine-induced auto tetraploid *Arabidopsis thaliana* [26], signifying the 45S rDNA-bearing chromosome reorganization, as an entity of intragenomic translocation, and in tetraploid *Centaurea jacea* [27] in which the deletion of one pair of 45S rDNA loci was observed. Unluckily, there was lack of concrete evidence about mechanism involved in rDNA loci variation, so further studies need to be done in order to find the reasons behind the variation in pattern of rDNA loci.

The observations recorded during the present investigation are slightly different as the polymorphism has been observed within the same loci which exist at the short arms of two

chromosomes of a homologous pair which seems like a rare observation, 'as it is well known fact that in case of variation in rDNA present in two chromosomes of a homologous pair it will be equilibrated by subsequent round of crossing over and recombination in next generations. Krishna is a variety of *C. flexuosus* which is developed from variety Pragati and Cauveri (*C. flexuosus)* through phenotypic recurrent selection programme [28]. The crosses led to homogenization of genes through recombination and crossing over hence there is no polymorphism in rDNA sites (Figs 1f and 2d) on both the homologous chromosomes and thus the intensity of rDNA hybridization signals remain similar in both chromosomes. However, Bio-13 (*C. winterianus*) was developed through mutation breeding. Manjari was developed by irradiation of the slips of variety Manjusha with gamma rays and Medini was developed by clonal selection of some of the well performing commercial varieties of *C. winterianus*. These facts prompted us to explore the possibilities for differential distribution of 45S rDNA in both chromosomes of a homologous chromosome pair existed in the above-mentioned varieties of *C. winterianus*.

Java citronella, i.e., *C. winterianus* flowers copiously in South India and at higher attitudes in the hills of North Eastern India. However, due to irregularities in meiosis and chromosome polyploidy, viable seeds are not formed and therefore, the species can be propagated only by vegetative means. As per perusal of different literature we come across the fact that heterochromatin sites are fragile in nature and are prone to degeneration upon the exposure of physical and chemical mutagens. Considering the importance of these species there are several programmes running in different institutions to increase quantity and quality aspect. The development of *C. winterianus* var. Manjari through gamma radiation may be responsible for breakage of fragile rDNA site from the chromosome or this variability may be due to evolutionary speciation due to natural cross among two different species of *Cymbopogon* which is fixed due to clonal propagation (Fig 3). However, in both the situations these changes were fixed by vegetative method of propagation which is general mode of reproduction in the case of *C. winterianus*. Thus, the varieties of *C. winterianus* never got chance to recover by chromosome homogenization. This may be the reason why the change got fixed in cells of these varieties of *C. winterianus*. This explanation was also found to be apt for the variety Medini of *C. winterianus* which is developed through clonal selection from Manjari and having the same background. The *C. winterianus* var. Bio-13 is developed by *in vitro* somaclonal selection [29]. However, seed setting in *C. flexuosus* var. Krishna takes place in all regions of India, and hence developed through cross pollination and recurrent selection. In addition to vegetative propagation, it is also being propagated through seeds, hence there is equal opportunity of chromosomal recovery in each generation.

## Material and methods

### Plant materials

Four varieties of two *Cymbopogon* species namely *C. winterianus* (Medini, Manjari, Bio-13) and *C. flexuosus* (Krishna) were collected from Central Institute of Medicinal and Aromatic Plants (CIMAP), Research Centre, Pantnagar, Uttarakhand, India. The information concerning the details of cultivars, chromosome number and percentage listed in Table 2.

### Preparation of chromosome spreads

The procedure of mitotic chromosome preparation was basically the same as published protocols [29] with some modifications. The slips of these cultivars were covered with moist paper and kept in a tray for root initiation in dark under room temperature for 68–72 hr. Lateral roots of about 1 cm were pre-treated in 0.002 mol/L 8-hydroxyquinoline at room temperature

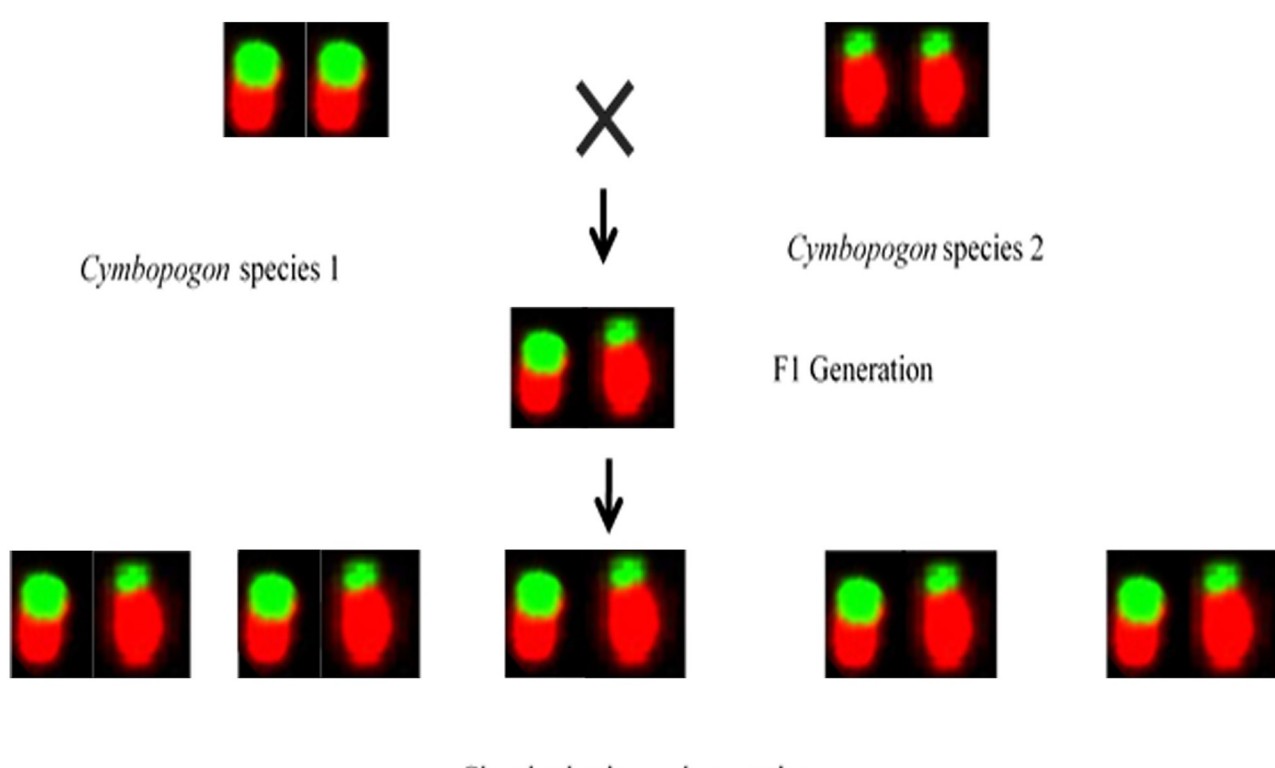

**Fig 3. Evolutionary speciation in *C. winterianus*.**

for 2 h, then fixed in 3:1 Carnoy's fixative solution for at least 1 day. To obtain the chromosome preparations, fixed root tips were digested with enzyme mixtures, containing 4% cellulose Onozuka R-10 (Merck, http://www.merck-chemicals.com) and 2.0% pectinase (Sigma-Aldrich, http://www.sigmaaldrich.com) in 1X PBS buffer, pH 5.5, at 37°C for 50 min. The enzyme solution was replaced by deionized water and kept on ice for at least 8 min. Following, the digested root tips were fixed in 3:1 Carnoy's fixative. The slides were obtained using a "flame-dried" method, according to the published protocol [30].

## Probe preparation

Plasmid pTa71 with a size 9.0 kb from wheat [31] was used for 45S rDNA sites. rDNA was labelled with Fluorescein-12-dUTP (ROCHE Diagnostics) using the nick translation method.

**Table 2. Chromosome number and parentage of *Cymbopogon* species used in the present study.**

| Species | Varieties | Chromosome number | Parentage/Development |
|---|---|---|---|
| *C. flexuosus* | Krishna | 20 | Phenotypic recurrent selection [28] |
| C. winterianus | Manjari | 20 | Induced Mutagenesis [32] |
| | Medini | 20 | Clonal selection [33] |
| | Bio-13 | 20 | *In vitro* somaclonal selection [29] |

## Signal detection and analysis

A 30 μl of hybridization mixture contained 15 μl of deionized formamide, 3 μl of 20X SSC, 6μl of 50% dextran sulphate, 1 μl of Salmon sperm DNA, 2 μl of probe DNA and 3 μl of ddH₂O was added to denatured slide and covered with plastic coverslip. The slides were then incubated at 37º C in moist chamber overnight for hybridization of labelled probe with target DNA. After overnight incubation, slides were washed three times in 2X SSC for 5 min, 50% formamide in 2X SSC for 10 min and three times with 2X SSC for 5 min each at 42º C in water bath. This was followed by subsequent washing in 1X SSC for 5 min at RT. The slides were counterstained with propidium iodide and mounted with single drop of anti-fade mounting medium (Vectashield) and covered with 22×30 mm coverslips.

Photographs of cells were captured with well spread chromosomes by epifluorescence Zeiss Axioimager MI microscope (Carl Zeiss, Germany). The mean length of each chromosome, chromosome length, long arm, short arm and arm ratio of each chromosome and percentage of 45S rDNA signals were obtained through measurements with MicroMeasure 3.3 software (http://www.colostate.edu/Depts/Biology/ MicroMeasure). The karyograms were developed using above parameters obtained through MicroMeasure 3.3.

## Acknowledgments

The author acknowledged the support of Central Institute of Medicinal and Aromatic Plants (CIMAP), Research Centre, Pantnagar, Uttarakhand, India for providing the germplasm. The research was performed at Molecular Cytogenetics Laboratory, Department of Molecular Biology & Genetic Engineering, College of Basic Sciences & Humanities, G. B. Pant University of Agriculture & Technology, Pantnagar, Uttarakhand.

## Author Contributions

**Data curation:** Shivangi Thakur, Reyazul Rouf Mir, Sundip Kumar.

**Formal analysis:** Shivangi Thakur, Rashmi Malik, Priyanka Balyan, Reyazul Rouf Mir, Sundip Kumar.

**Investigation:** Upendra Kumar, Priyanka Balyan, Sundip Kumar.

**Methodology:** Upendra Kumar, Darshana Bisht.

**Project administration:** Sundip Kumar.

**Supervision:** Upendra Kumar.

**Validation:** Shivangi Thakur.

**Writing – review & editing:** Reyazul Rouf Mir, Sundip Kumar.

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
