## [Decision Letter · Decision Letter 0]

20 Sep 2021

PONE-D-21-26873Physical localization of 45S rDNA in Cymbopogon and the analysis of differential distribution of rDNA in homologous chromosomes of Cymbopogon winterianus.PLOS ONE

Dear Dr. KUMAR,

Thank you for submitting your manuscript to PLOS ONE. After careful consideration, we feel that it has merit but does not fully meet PLOS ONE’s publication criteria as it currently stands. Therefore, we invite you to submit a revised version of the manuscript that addresses the points raised during the review process.

We look forward to receiving your revised manuscript.

Kind regards,

Dengcai Liu, PhD

Academic Editor

PLOS ONE

Journal Requirements:

Reviewers' comments:

Reviewer's Responses to Questions

**Comments to the Author**

1. Is the manuscript technically sound, and do the data support the conclusions?

Reviewer #1: Yes

Reviewer #2: Yes

Reviewer #3: Yes

2. Has the statistical analysis been performed appropriately and rigorously? 

Reviewer #1: No

Reviewer #2: N/A

Reviewer #3: Yes

3. Have the authors made all data underlying the findings in their manuscript fully available?

Reviewer #1: No

Reviewer #2: Yes

Reviewer #3: Yes

4. Is the manuscript presented in an intelligible fashion and written in standard English?

Reviewer #1: Yes

Reviewer #2: Yes

Reviewer #3: Yes

5. Review Comments to the Author

Reviewer #1: The karyotypes of Cymbopogon species were constructed based on arm ratios and the physical localization of 45S rDNA in Cymbopogon was also determined. The research content of this article is too simple. Although the 45S rDNA was located on a pair of chromosomes and the signal diversity was observed, these results can not display the evolution of Cymbopogon species, because only four varieties were analyzed. The novelty and scientific significance of this manuscript are limited. Additionally, the authors didn't write the article carefully. For example, Table 2 can not be found in the main text. It was not indicated how many cells for each material were detected, and the statistical data was not listed.

Reviewer #2: The manuscript reported localization and genetic variation of 45S rDNA in representative Cymbopogon species. The study was clearly presented, with conclusions being supported by multiple evidence. The results were quite interesting for classification, genetic diversity and evolutionary studies of Cymbopogon.

This manuscript should be suitable for publication after replying below questions and considering minor revision.

1. The first time of scientific name for genus Cymbopogon, and after that, the Cymbopogon may be abbreviated by C. for all species in the genus, which may be checked overall manuscript.

2. The quality of “Figures” was well, however, the scale bar for the figures should be indicated.

3. For the chromosome lengths and strength of rDNA FISH signals in the species, the statistic number of cells may be mentioned for each samples of the species.

4. The second passage in “Discussion” will be condensed to present the knowledge closely associate to the present studies.

5. The citation style of “References” should be checked in detail.

Reviewer #3: The manuscript entitled “Physical localisation of 45S rDNA in Cymbopogon and the analysis of differential distribution of rDNA in homologous chromosome of Cymbopogon winterianus” by Thakur et al. is a very timely research article. The results are interesting with good merit. The authors provided new cytogenetic data for several cultivars of a Lemon grass/Cymbopogon. 45S rDNA sites which were used as probes for performing FISH analysis gave considerable interspecific variation in the intensity of 45S rDNA hybridization signals in both the cultivars. This helps in distinguishing somatic chromosomes clearly. Secondly, it yields information about differential distribution of 45S rDNA hybridization signals in heterologous chromosomes which further gave insight in evolutionary differences in different cultivars of the Cymbopogon. This shall provide immense helps in designing economic plant breeding strategies for this crop for further improvement. However, the manuscript needs following changes before recommending for the publication.

1. The language and readability of the manuscript needs some improvement.

2. Works have also be done earlier on molecular marker development and their use in diversity analysis in this species. It should be reviewed properly in introduction. For example, Kumar, J, Verma V, Qazi GN and Balyan HS (2007). Developments of simple sequence repeat markers in Cymbopogon species. Planta Medica 73: 262-266 Kumar J , Verma V, Qazi GN and Gupta PK (2007) Genetic diversity in Cymbopogon species using PCR-based functional markers. Journal of Plant Biochemistry and Biotechnology 16(2): 119-122

3. The presentation of data in figure needs improvement as they were overly stretched (Figure 1) and lacking scales.

4. The presentation of data in results section demands improvement

5. Like results section, the discussion section may be divided into different sub sections.

6. Line 105-111: Reframe the sentences.

7. Line 106, 113, 120: which indicated instead of which is indicating.

8. Line 133-137: Rephrase the sentence.

9. Line 145-150: Elaborate the sentence properly.

10. Line 148: Space between triggering and relocation.

11. Line 80-83: Rewrite the sentence.

12. There is repetition in line 81-82 and 88-89.

13. Line 145: the extensive studies instead of the extensive study.

14. Line 195-196: sounds incomplete, reframe it.

15. Line 214: space between 2 and %.

6. PLOS authors have the option to publish the peer review history of their article (what does this mean?). If published, this will include your full peer review and any attached files.

Reviewer #1: No

Reviewer #2: No

Reviewer #3: **Yes: **Jitendra Kumar

---

## [Author Response · Author response to Decision Letter 0]

6 Oct 2021

Point wise response to comments of the Reviewer#1

The karyotypes of Cymbopogon species were constructed based on arm ratios and the physical localization of 45S rDNA in Cymbopogon was also determined. The research content of this article is too simple. Although the 45S rDNA was located on a pair of chromosomes and the signal diversity was observed, these results cannot display the evolution of Cymbopogon species, because only four varieties were analyzed. The novelty and scientific significance of this manuscript are limited. Additionally, the authors didn't write the article carefully. For example, Table 2 cannot be found in the main text. It was not indicated how many cells for each material were detected, and the statistical data was not listed.

Response: Many thanks for spending quality time and reviewing our manuscript which helped to improve the overall quality of this manuscript. We have taken your comments into consideration while revising our manuscript. In response to your comments we would like to state that: (i) The research content although seems simple, but we were successful in deriving important information related to evolution of Cymbopogon species which can prove useful in developing advanced plant breeding programmes in future for this crop species. (ii) In response to your comment regarding the number of genotypes used, we would like to state that only four varieties of Cymbopogon are actually available in India. However, we would like to work on more varieties in near future if more genotypes will become available. (iii) The karyotyping and localization of 45S rDNA was carried out first time on this species. (iv) The whole manuscript has been thoroughly revised and changes have been made throughout the manuscript to increase its readability. The languages of the manuscript has been also improved. (v) We agree there was a typing mistake in citing Table #2. Table 2 was found in material and method section. (vi) The number of cells in each material is mentioned in Table 1.

Point wise response to comments of the Reviewer#2

Reviewer #2: The manuscript reported localization and genetic variation of 45S rDNA in representative Cymbopogon species. The study was clearly presented, with conclusions being supported by multiple evidence. The results were quite interesting for classification, genetic diversity and evolutionary studies of Cymbopogon.

This manuscript should be suitable for publication after replying below questions and considering minor revision.

Response: Many thanks for spending quality time, reviewing and appreciating our work.

1. The first time of scientific name for genus Cymbopogon, and after that, the Cymbopogon may be abbreviated by C. for all species in the genus, which may be checked overall manuscript.

Response: As desired, needful has been done.

2. The quality of ‘Figures’ was well, however, the scale bar for the figures should be indicated

Response: Needful has been done by providing scalebar for the figures.

3. For the chromosome lengths and strengths of rDNA FISH signals in the species, the statistic number of cells may be mentioned for each samples of the species.

Response: The number of cells studied for each species mentioned in Table 1.

4. The second passage in “Discussion” will be condensed to present the knowledge closely associate to the present studies.

Response: As desired, needful has been done.

5. The citation style of “References” should be checked in detail.

Response: As desired, needful has been done.

Point wise response to comments of the Reviewer#3

Reviewer #3: The manuscript entitled “Physical localisation of 45S rDNA in Cymbopogon and the analysis of differential distribution of rDNA in homologous chromosome of Cymbopogon winterianus” by Thakur et al. is a very timely research article. The results are interesting with good merit. The authors provided new cytogenetic data for several cultivars of a Lemon grass/Cymbopogon. 45S rDNA sites which were used as probes for performing FISH analysis gave considerable interspecific variation in the intensity of 45S rDNA hybridization signals in both the cultivars. This helps in distinguishing somatic chromosomes clearly. Secondly, it yields information about differential distribution of 45S rDNA hybridization signals in heterologous chromosomes which further gave insight in evolutionary differences in different cultivars of the Cymbopogon. This shall provide immense helps in designing economic plant breeding strategies for this crop for further improvement. However, the manuscript needs following changes before recommending for the publication.

Response: Many thanks for spending quality time and reviewing our manuscript. Many thanks also for appreciating our work.

1. The language and readability of the manuscript needs some improvement.

Response: The whole manuscript has been thoroughly revised and changes have been made throughout the manuscript. The languages and readability of the manuscript has been also improved.

2. Works have also be done earlier on molecular marker development and their use in diversity analysis in this species. It should be reviewed properly in introduction. For example, Kumar, J, Verma V, Qazi GN and Balyan HS (2007). Developments of simple sequence repeat markers in Cymbopogon species. Planta Medica 73: 262-266 Kumar J , Verma V, Qazi GN and Gupta PK (2007) Genetic diversity in Cymbopogon species using PCR-based functional markers. Journal of Plant Biochemistry and Biotechnology 16(2): 119-122.

Response: As desired, needful has been done

3. The presentation of data in figure needs improvement as they were overly stretched (Figure 1) and lacking scales.

Response: As desired, needful has been done.

4. The presentation of data in results section demands improvement

Response: As desired, needful has been done.

5. Like results section, the discussion section may be divided into different sub sections.

Response: It was not possible, although for perusal of the viewers, discussions was divided into different paragraphs.

6. Line 105-111: Reframe the sentences.

Response: As desired, needful has been done.

7. Line 106, 113, 120: which indicated instead of which is indicating.

Response: As desired, needful has been done.

8. Line 133-137: Rephrase the sentence.

Response: As desired, needful has been done.

9. Line 145-150: Elaborate the sentence properly.

Response: As desired, needful has been done.

10. Line 148: Space between triggering and relocation.

Response: As desired, needful has been done.

11. Line 80-83: Rewrite the sentence.

Response: As desired, needful has been done.

12. There is repetition in line 81-82 and 88-89.

Response: As desired, needful has been done.

13. Line 145: the extensive studies instead of the extensive study.

Response: As desired, needful has been done.

14. Line 195-196: sounds incomplete, reframe it.

Response: As desired, needful has been done.

15. Line 214: space between 2 and %.

Response: As desired, needful has been done.

---

## [Editor Report · Decision Letter 1]

8 Oct 2021

Physical localization of 45S rDNA in Cymbopogon and the analysis of differential distribution of rDNA in homologous chromosomes of Cymbopogon winterianus.

PONE-D-21-26873R1

Dear Dr. KUMAR,

We’re pleased to inform you that your manuscript has been judged scientifically suitable for publication and will be formally accepted for publication once it meets all outstanding technical requirements.

Kind regards,

Dengcai Liu, PhD

Academic Editor

PLOS ONE

Additional Editor Comments (optional):

Thank your improvement on the manuscirpt.

---

## [Editor Report · Acceptance letter]

14 Oct 2021

PONE-D-21-26873R1 

Physical localization of 45S rDNA in *Cymbopogon* and the analysis of differential distribution of rDNA in homologous chromosomes of *Cymbopogon winterianus*

Dear Dr. Kumar:

I'm pleased to inform you that your manuscript has been deemed suitable for publication in PLOS ONE. Congratulations! Your manuscript is now with our production department. 

Kind regards, 

on behalf of

Dr. Dengcai Liu 

Academic Editor

PLOS ONE